



# The ALOMAR Rayleigh / Mie / Raman lidar: status after 30 years of operation

Jens Fiedler[1] and Gerd Baumgarten[1]

[1]Leibniz Institute of Atmospheric Physics at the University of Rostock, Kühlungsborn, Germany

**Correspondence:** Jens Fiedler (fiedler@iap-kborn.de)

**Abstract.**

The ALOMAR Rayleigh/Mie/Raman (RMR) lidar is an active remote sensing instrument for investigation of the Arctic middle atmosphere on a routine basis during day and night. It was installed on the island of Andøya in Northern Norway (69° N, 16° E) in summer 1994. During the past 30 years of operation, more than 20,200 hours of atmospheric data were measured, approx. 60 % thereof during sunlit conditions. At present, the RMR lidar is the only system measuring aerosols, temperature, and horizontal winds simultaneously and during daytime in the middle atmosphere. We report on the current status of the lidar, including major upgrades made during recent years. This involves a new generation of power lasers and new systems for synchronization, data acquisition, and spectral monitoring of each single laser pulse. Lidar measurements benefit significantly from a control system for augmented operation with automated rule-based decisions, which allows complete remote operation of the lidar. This was necessary in particular during the COVID-19-pandemic, as it was impossible to access the lidar from outside Norway for almost 1.5 years. We show examples that illustrate the performance of the RMR lidar in investigating aerosol layers, temperature as well as horizontal winds, partly with a time resolution below one second.

## 1   Introduction

The idea of using electric light for exploring the atmosphere was contemplated already more than 130 years ago. Jesse (1887) proposed to "shoot a bundle of intensive collimated electric light" in the direction of high altitude clouds to determine the altitude of so-called noctilucent clouds (NLC) which form in summer at mid- and high latitudes in approximately 83 km above Earth's surface. The first implementation of this idea was around 1950, using a high-voltage spark between aluminum electrodes as light source, two searchlight mirrors as the transmitter and receiver optics, and a photoelectrical cell as detector. This system successfully measured cloud-base heights up to 5.5 km in bright daylight (Fujii and Fukuchi, 2005). Later on, such systems, consisting of a light source and a time-resolving detection system, were given the acronym LIDAR (LIght Detection And Ranging) just as RADAR describes instruments using radio waves as a radiation source (Middleton and Spilhaus, 1953). The principle of a lidar is rather simple and based on the transmission of a short light pulse into the air and the detection and analysis of scattered radiation. Scattering occurs on solid and liquid objects as well as air molecules, certainly with different efficiency. The distance between transmitter and scattering body can be determined using the transit time of the light between these two locations. While the principle was used before, the development of powerful lidar systems was strongly pushed



by the invention of the LASER (Light Amplification by Stimulated Emission of Radiation) in 1960 (Maiman, 1960). Laser instruments are coherent light sources and basically consist of a gain medium which is able to produce a population inversion, a pump source to bring energy into the gain medium for the amplification process, and a resonator around the gain medium. The resonator enhances the output power, collimates the laser beam and provides spectral purity of the outgoing radiation. There might be used additional components, e.g. Q-switches, for producing short pulses with high power.

Lidars for exploring the middle atmosphere cover an extensive altitude range up to around 100 km, where the air density decreases by about six orders of magnitude. As the received light is mostly generated by elastic scattering at air molecules, the detected signal decreases exponentially with altitude. For this reason, middle atmosphere lidars are mostly based on high-power solid-state lasers to extend the measurement range as high as possible. Among such lasers, the Nd:YAG laser is the most commonly used. Its gain medium (neodymium-doped yttrium aluminum garnet) has been applied since 1964 for lasers which are widely used in industrial, medical and scientific applications. While the fundamental wavelength is in the infrared, middle atmosphere lidars use most often the frequency doubled wavelength at 532 nm, e.g., the systems at Arecibo Observatory / Puerto Rico (18° N, 67° W) (Tepley et al., 1993), Maïdo Observatory / France (21° S, 55° E) (Baray et al., 2013), Wuhan / China (30° N, 114° E) (Chang et al., 2005), Delaware Observatory / Canada (43° N, 81° W) (Sica et al., 1995), Observatory of Haute-Provence / France (44° N, 6° E) (Khaykin et al., 2019), IAP Kühlungsborn / Germany (54° N, 11° E) (Gerding et al., 2016), and the DLR / Germany mobile lidars (Kaifler et al., 2017; Kaifler and Kaifler, 2021).

Measurements in Arctic regions are particularly challenging as the sun is always above the horizon during summer. It is essential to suppress the broad-band spectrum of scattered solar radiation significantly to extend the measurements over as many hours of the day as possible. This is commonly achieved by a reduction of the telescope field-of-view, as well as the usage of narrow-bandwidth optical filters in the receiver. Such systems are usually called "daylight capable". Middle atmosphere lidars which have been operated at Arctic locations for several years are at, e.g., Poker Flat / USA (65° N, 147° W) (Cutler et al., 2001), Sondrestrom Facility / Denmark (67° N, 51° W) (Thayer et al., 1997), Esrange Space Center / Sweden (68° N, 21° E) (Blum and Fricke, 2005), Davis Station / Australia (69° S, 78° E) (Klekociuk et al., 2003), Syowa Station / Japan (69° S, 40° E) (Kogure et al., 2017), ALOMAR / Norway (69° N, 16° E) (von Zahn et al., 2000), and Eureka Observatory / Canada (80° N, 86° W) (Duck et al., 1998). Only some of these lidars were able to measure during sunlit conditions.

The Rayleigh/Mie/Raman (RMR) lidar at the Arctic Lidar Observatory for Middle Atmosphere Research (ALOMAR) is located on the island of Andøya in Northern Norway (69° N, 16° E). It is designed for multi-parameter investigations of the Arctic middle atmosphere on a climatological basis and was during the initial period a joint effort between four European partners: the Leibniz-Institute of Atmospheric Physics (Kühlungsborn, Germany), the Institute of Physics of Bonn University (Bonn, Germany), the Service d'Aéronomie du C.N.R.S. (Verrières le Buisson, France) and Hovemere Ltd. (Keston, UK) (von Zahn et al., 2000). The RMR lidar is a complex twin-system consisting of two power lasers each emitting at three wavelengths simultaneously, two steerable telescopes, numerous detection channels for different wavelengths and altitude ranges, and narrowband daylight filters. Although the lidar is generally used for routine soundings of the middle atmosphere, it is additionally operated during special measurement campaigns with the help of external students. The RMR lidar is in operation since mid-1994 and collected more than 20,200 hours of atmospheric data since then. After significant technical



upgrades, the DORIS (Doppler Rayleigh Iodine Spectrometer) was integrated into the lidar which upgraded the system to a direct detection Doppler lidar (Baumgarten, 2010). Thereby, the RMR lidar is capable of measuring fundamental parameters of the middle atmosphere simultaneously, like stratospheric aerosols (e.g., Gerding et al., 2003; Langenbach et al., 2019), mesospheric ice particles (e.g., von Cossart et al., 1997; Baumgarten et al., 2002; Fiedler et al., 2011), temperature (e.g., von Zahn et al., 1998; Schöch et al., 2008; Hildebrand et al., 2017), and wind (e.g., Hildebrand et al., 2012; Baumgarten et al., 2015).

Since the installation, the technical performance of the lidar has been continuously improved. Major activities are scattered in the literature and can be found in Rees et al. (1996), Fiedler and von Cossart (1999), von Zahn et al. (2000), Fiedler et al. (2008), Baumgarten (2010), Fiedler and Baumgarten (2012), and Fiedler et al. (2017). During the last few years, the lidar was significantly upgraded, which enables new possibilities for sounding the middle atmosphere using this system. In the next sections, we will report the status of the RMR lidar and give examples of atmospheric measurements showing the performance of the system.

## 2 System setup

The RMR lidar was installed in spring 1994 as the first geophysical instrument in the ALOMAR building. Its components are distributed over several rooms of the two floors of the building. Figure 1 shows the block diagram of the current lidar setup. The system consists of two identical laser transmitters (TMT1, TMT2) and telescopes (NWT, SET) but only one polychromatic detection system (PCD). Each laser transmitter itself is a complex subsystem, which is optically driven by a frequency stabilized seeder (SDR) and described in section 2.1. The outgoing beams of TMT1 and TMT2 contain three wavelengths, have a diameter of 200 mm and are guided by three computer controlled mirrors into the atmosphere, respectively. The mappings between laser transmitters and telescopes are flexible and set by the beam guiding mirrors. Both telescopes are independently steerable from zenith pointing to 30° off-zenith. Their orientation is such that one telescope (NWT) can be tilted to the north-west quadrant and the other one (SET) to the south-east quadrant (section 2.2). This setup allows, e.g., simultaneous measurements of zonal and meridional wind components or momentum fluxes. Light scattered back from air molecules and particles, as well as solar background, is collected by the 1.8 m diameter primary telescope mirrors and guided via multimode fibers to a fiber switch. Here, the light from three sources (NWT, SET, SDR) is coupled sequentially into the PCD where it is separated according to wavelength and intensity into a number of detection channels. After the conversion of optical to electrical intensities by avalanche photodiodes or photomultiplier tubes, the signals are analyzed by data acquisition systems (DAQ), see section 2.3.

All components of the lidar are synchronized by a dedicated trigger controller (TRG), which provides the correct timing for the two alternating working transmitters. The absolute lidar system time is driven by a Global Positioning System (GPS) satellite receiver. A central system status is collected and delivered by a message queuing telemetry transport (MQTT) server. More details regarding synchronization follow in section 2.4.



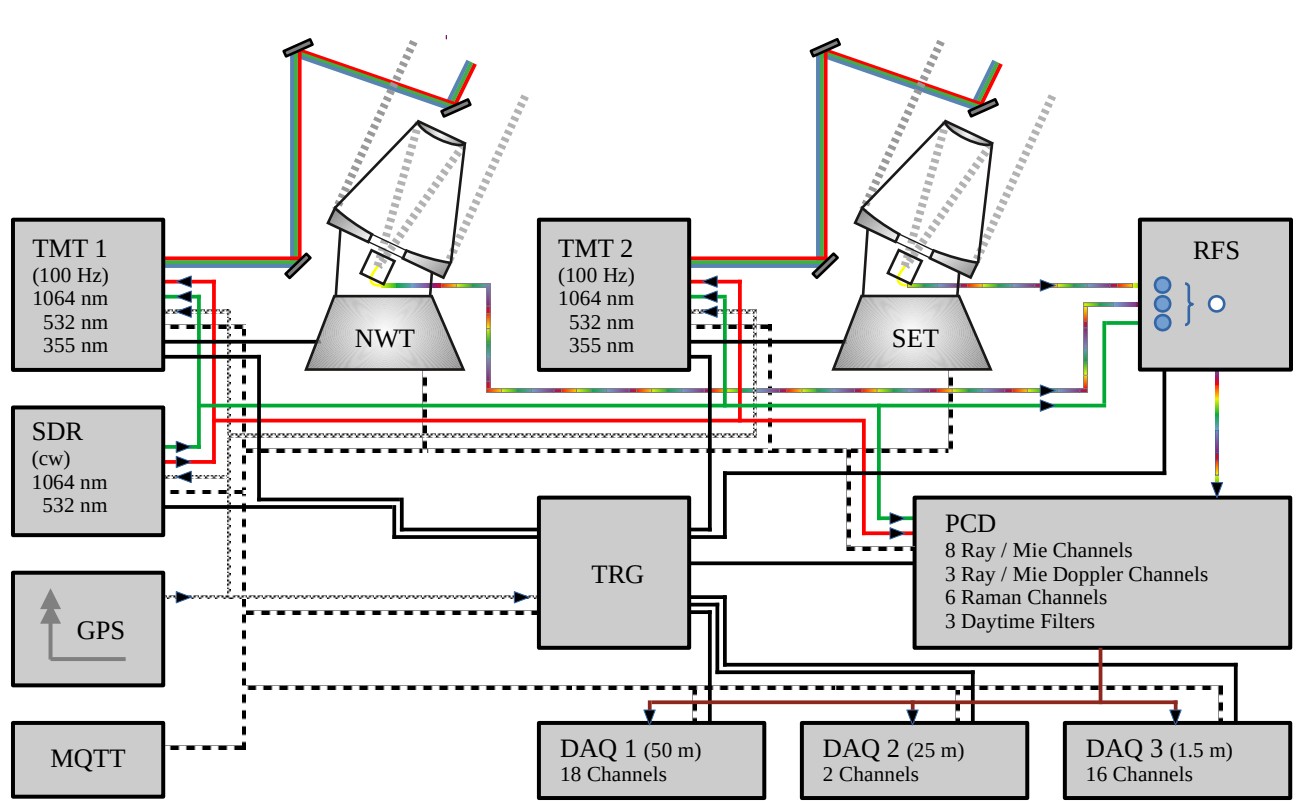

**Figure 1.** Overview of the ALOMAR RMR lidar: laser transmitters (TMT1, TMT2) are optically driven by a seeder (SDR) and emit three wavelengths simultaneously at 100 Hz repetition rate each. Telescopes collect light in the north-west (NWT) and south-east (SET) quadrants. A polychromatic detection system (PCD) splits received light by wavelength and intensity. Data acquisition systems (DAQ1, DAQ2, DAQ3) analyze the signals, a trigger controller (TRG) synchronizes the lidar components, a satellite clock (GPS) provides the absolute system time, and a lightweight messaging protocol server (MQTT) distributes the lidar status. Red and green lines indicate light from the seed laser, which is distributed to other systems. The different styles of black lines indicate various types of control signals.

## 2.1 Transmitter

The RMR lidar is driven by a set of two Nd:YAG power lasers. Each laser system has an identical set-up on an optical table, together with equipment for beam guiding, expansion, and analysis. Figure 2 shows the block diagram for one laser system.

For a long time, flashlamp pumped Nd:YAG lasers from Spectra-Physics were used, the GCR-6-30 model from 1994 to 2003 and the PRO-290-30 model from 2003 to 2018. Its flashlamps had life times of approximately $50 \times 10^6$ pulses, which caused up to three maintenances stays at the lidar location per year for their exchange and readjustment work. Currently, the third set of power lasers is in operation. The EVO-IV lasers from Innolas GmbH are diode pumped Nd:YAG lasers which have the potential of long maintenance free operation, as the pump diodes are expected to last for more than $2 \times 10^9$ pulses. This would

convert to approximately seven years of operation, considering the mean lidar measurement time during the past years. The





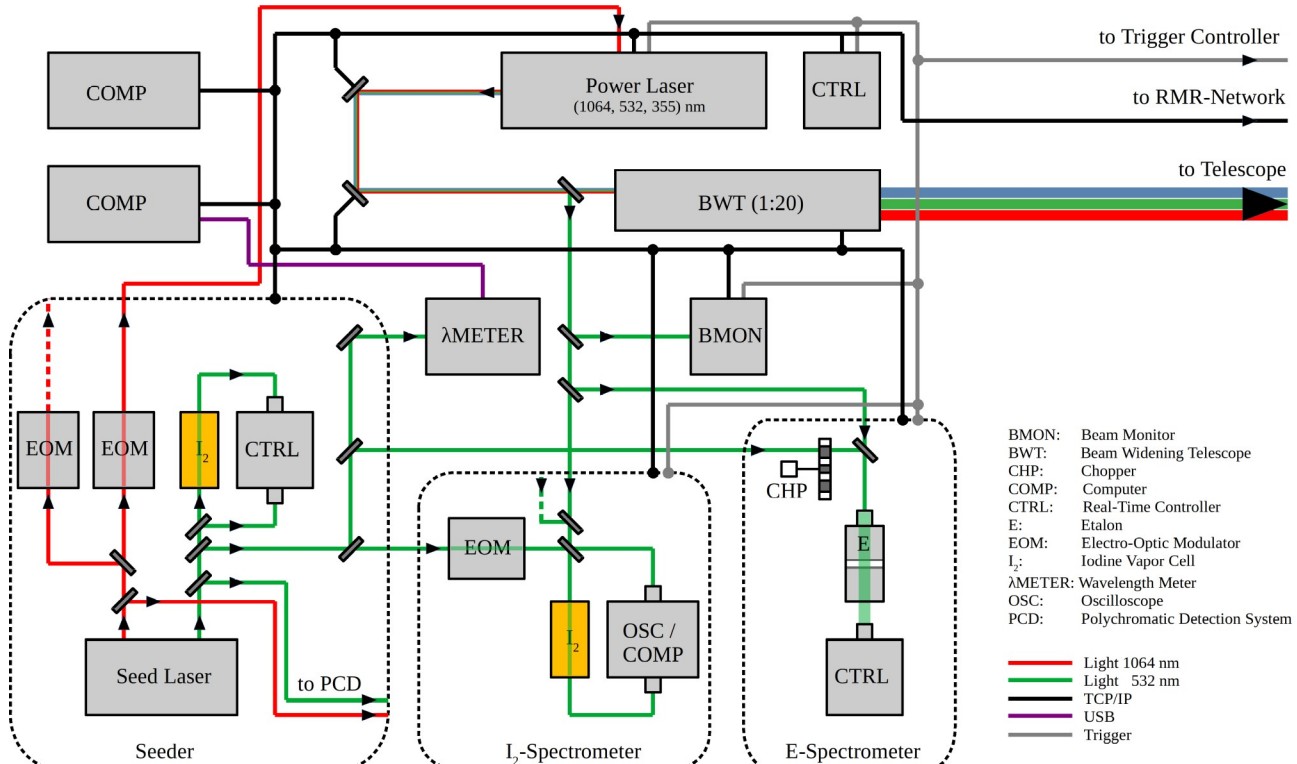

**Figure 2.** Overview of the ALOMAR RMR lidar transmitter. The power laser setup is only shown for one of the two (identical) systems. Dashed red (green) lines go to (come from) the second system. The laser beam, containing three wavelengths, is guided by two steerable mirrors and its diameter is increased by a factor of 20 using a beam widening telescope. The seeder unit generates light at 1064 nm and 532 nm, which is frequency stabilized by iodine absorption spectroscopy, and used for injection seeding of the power lasers as well as analysis and calibration purposes. A wavelength meter and two different types of spectrometers provide information regarding the spectral properties of the outgoing laser pulses. Dashed frames group components belonging to the indicated devices. For details, see text.

lasers can be entirely remote controlled. They also provide the possibility for optimized output at either all three or only two wavelengths, which can be set by computer control.

The primary laser wavelength of 1064 nm is frequency doubled to 532 nm and tripled to 355 nm by nonlinear crystals and the outgoing beam, containing all three wavelengths, is guided via two tilted mirrors to a beam widening telescope. Both mirrors are coated for high reflection on the laser wavelengths and mounted on solid-state joints with piezo drivers, allowing an absolute angle accuracy of 1 μrad. The mirror mounts are computer controlled and form, together with a CCD camera and a computer, a closed loop which allows a direction stabilization of the laser beam before entering the beam widening telescope. The telescope is an off-axis collimator having a focal length of 2000 mm which is mounted in a tube of 2040 mm length and 350 mm diameter. Here, the diameter of the laser beam (~9 mm) is widened by a factor of 20 before transmission into the atmosphere. The collimation quality depends on the distance and angles of the two telescope mirrors, which have to be adjusted





using a complex, time-consuming autocollimation procedure. To simplify the process and gain more flexibility, the original mount of the spherical mirror, which was in use since 1994, has been exchanged recently with a piezo motor-driven OEM actuator by Physik Instrumente GmbH. This device has a linear travel range of 14 mm and features three degrees of freedom for tilting. The linear and angular resolutions are 500 nm and 150 μrad, respectively, which allows precise and reproducible

adjustments of the telescope parameters. Moreover, these computer controlled settings offer the possibility of changing the beam focus in the atmosphere by moving of the spherical mirror relative to the autocollimation position.

To operate both power lasers at the same wavelength, they are injection seeded by an external cw Nd:YAG laser (Innolight Prometheus-100NE). This single-mode seed laser generates both the fundamental and second harmonic wavelengths, which are phase-locked to each other. Additionally, the laser provides frequency tuning, both slow over a larger range via the laser

crystal temperature and fast over a narrow range via a piezo element on the laser crystal. Both parameters can be controlled with external voltages applied to the laser head, which is used for an absolute frequency stabilization of the seed laser output by iodine absorption spectroscopy. For that, 532 nm radiation is guided through an iodine vapor cell and the light intensity is determined before and behind the cell using thermally stable photodiodes. The ratio of both intensities is a measure of the absorption and depends on the spectral properties of the gas ($I_2$) and the laser wavelength. The thermal tuning range of the seed

laser is ~60 GHz which groups into several ~14 GHz continuous tuning ranges between mode-hops and is sufficiently large to cover a number of iodine absorption lines. The piezo tuning range of ~400 MHz is used to lock the laser to the slope of $I_2$-line 1109. For low temperate drift of the Doppler broadened $I_2$-line the absorption cell is heated and stabilized to 40° C. At this temperature, the iodine is completely in the gas phase, resulting in a temperature dependence of only 0.7 MHz / K. The photodiodes, measuring the light intensities, and the laser frequency control inputs are connected to an embedded controller (National

Instruments cRIO-9035). This class of controllers features a real-time operating system (RT-Linux), field programmable gate array (FPGA), and reconfigurable input/output hardware (RIO). Both, RT and FPGA level, can be programmed with high-level languages like the graphical programming system LabVIEW. The controller acts as a stand-alone device and stabilizes the seed laser frequency at a rate of ~200 Hz to better than $\Delta\lambda / \lambda = 10^{-8}$. The software interacts with other computers via TCP/IP networking.

Radiation at 1064 nm is guided via single-mode optical fibers from the seed laser to both power lasers, where it is coupled into the laser cavity for injection seeding. This technique is used to obtain pulses of high power with a narrow bandwidth. For this purpose, the frequency of the seed light has to be close to the resonance frequency of one particular resonator mode of the power laser oscillator. We realize this by the standard method, called "pulse build-up time reduction", but use our own custom setup. The basic principle relies on the measurement of the time delay between the opening of the Pockels cell (Q-switch) and

the outgoing laser pulse of the oscillator. This delay is in the order of 140 ns for our power lasers and shows lowest values when seed light frequency and resonator length match to each other. One of the oscillator mirrors is mounted on a piezo phase shifter with 3 μm displacement range, whose position determines the oscillator length. Both, build-up time measurement and phase shifter steering, are performed by an embedded controller (National Instruments sbRIO-9637). This single-board OEM device has similar properties to the one used for controlling the seed laser and is integrated into a 19" box (laser controller),

which additionally holds electronics for signal conditioning and time measurements. The software running on the embedded





**Table 1.** Basic parameters of the ALOMAR RMR lidar transmitter, optimized for 532 nm output.

| Seed laser: Single-mode cw Nd:YAG (Innolight Prometheus-100NE) | |
| --- | --- |
| Parameter | Value |
| Power (max) at 1064 nm | 1600 mW |
| Power (max) at 532 nm | 105 mW |
| Spectral stability $\Delta\lambda / \lambda$ at 532 nm | $< 10^{-8}$ |
| Power lasers: 2 x Injection seeded, pulsed Nd:YAG (Innolas Spitlight DPSS EVO-IV) | |
| Parameter | Value |
| Pulse energy at 1064 nm | ∼185 mJ |
| Pulse energy at 532 nm | ∼465 mJ |
| Pulse energy at 355 nm | ∼150 mJ |
| Pulse length at 532 nm | ∼9 ns |
| Repetition rate | 100 Hz |
| Beam diameter after expansion | 180 mm |
| Beam divergence after expansion | 25 μrad |

controller executes the minimization of the pulse build-up time and controls the power laser (status, trigger). To avoid cw radiation at 1064 nm from the seed laser to be detected by the lidar receiver, it is blocked using an electro-optic modulator (EOM) shortly after the laser pulse. Table 1 shows the basic parameters of the lidar transmitter.

## 2.2 Telescopes and beam guiding

Both telescopes and the mirrors for guiding the 180 mm diameter laser beams are installed in a telescope hall of $7 \times 7 \times 7$ m dimension. The telescopes are Cassegrain systems, having ultra-low coefficient of thermal expansion glass-ceramic (Sitall) primary mirrors of 1.8 m diameter, secondary mirrors of 0.58 m diameter, a focal length of 8.34 m, a numerical aperture of 0.11 and a field-of-view of ∼100 μrad. The focal plane is within a box mounted underneath the primary mirror which contains additional optics, motorized optical fibers holders and a fast camera. Both telescopes are fully motorized and can change

viewing directions within one quadrant of the sky, namely from north to west (NWT) and from south to east (SET). The pointing can vary between zenith and 30° off-zenith.

The laser beams enter the telescope hall horizontally through holes in the wall, are deflected vertically by the first set of beam guiding mirrors to the top of the telescope hall and are distributed by the second set of mirrors slightly downward to the top of the telescopes. Here, the third set of mirrors is mounted on top of the secondary mirror assemblies of the telescopes to

align the laser beams with the viewing directions of the telescopes for coaxial transmission, see also schematic in Figure 1. All

mirrors are motorized and computer controlled. The mappings between lasers and telescopes are flexible and determined by the alignment of the second and third set of mirrors.

The light scattered back from the atmosphere at around 1 km altitude is monitored by a fast, gated camera and the mirror on top of the telescope is automatically aligned to keep the laser beam within the telescope field-of-view. The images of the laser

are acquired for every laser pulse, and analyzed to find the center of mass. Even with this active control of the laser pointing, we still see remaining beam fluctuations in the order of 6 to 20 μrad corresponding to a miss-pointing of only 0.6 to 2 m at 100 km distance.

The initial design of the telescopes was to achieve full overlap at altitudes above 15 km (Baumgarten, 2001). Until 2015 the light below about 10 km was blocked by a mechanical chopper on the polychromatic detection system (von Zahn et al.,

2000). Since recent upgrades of the detection system (see below) also data from altitudes below 10 km is recorded. A detailed investigation about the overlap function of the telescopes is in progress.

### 2.3 Polychromatic detection and data acquisition

Light collected by the receiving telescopes is guided via optical fibers of 0.8 mm diameter to a polychromatic detection system (PCD). To simplify the instrumental setup, a single PCD is used for both laser/telescope systems. For that reason, the three

light sources (Seeder, NWT, SET) are coupled alternating into the detection system. This is realized by a fast-moving mirror (Galvano scanner), which acts as a rapid fiber selector (RFS). The RFS is synchronized to the pulses of one of the power lasers, and the mirror passes through a sequence of 3 positions. These positions have to be reproduced with high accuracy for each laser pulse in order to inject each light source into the optical axis of the detection system. The mirror remains for 0.5 ms at the seeder position and for 1.5 ms at each telescope position, which results in an upper altitude limit of 225 km for light scattered

back from the atmosphere.

After beam collimation the light is separated into different wavelength branches by dichroic mirrors. Basically, light from all three outgoing wavelengths (1064, 532, 355 nm) and several Raman shifted wavelengths is analyzed. This includes vibrational-Raman scattered photons at $N_2$ (387 nm and 608 nm) and $H_2O$ (660 nm). For the most intense outgoing radiation, the second harmonic wavelength of the lasers, rotational-Raman scattered photons at 529 nm and 530 nm are separated. For all branches

measuring at the outgoing wavelengths, narrowband interference filters are used, which reduces the "Rayleigh" scattered signal in these branches to the pure Cabannes line. A schematic is shown in Figure 3 and Table 2 provides details regarding the wavelengths channels. Fabry-Pérot etalons having bandwidths between 4 pm and 10 pm allow atmospheric measurements during daylight at these wavelengths, even during the highest solar elevation angles. The etalons for 1064 nm and 355 nm work with piezo actuators and capacitance stabilization, whereas the etalon for 532 nm is tuned by varying the pressure.

During nighttime, the etalons for 355 nm and 532 nm are bypassed by motorized flip mirrors FM, causing higher transmissions in these branches, and allowing for monitoring the transmission of the etalons. For the same purpose, the filter D3 can be removed during daytime. To account for the dynamic range of the received photons with altitude, several wavelength branches are intensity cascaded by beam splitters in up to three different channels before photo detection. In addition to atmospheric light, the 532 nm branch contains light from the seed laser for adjustment and monitoring purposes of the etalon filter and the





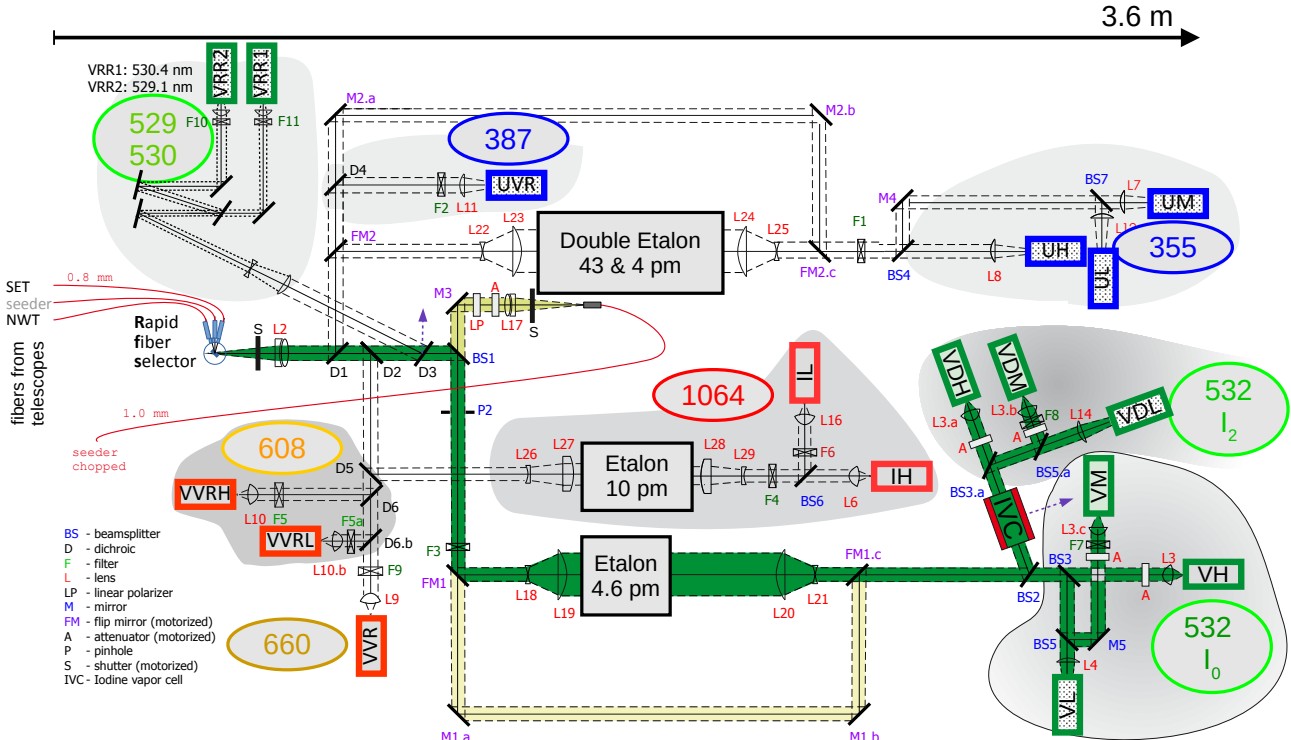

**Figure 3.** Polychromatic detection system of the ALOMAR RMR lidar. Light from the telescopes (NWT, SET) is fed alternating into the detection system using a fiber switch (rapid fiber selector). It is then split by wavelength and intensity. Light for the DORIS system is detected by the channel groups 532-$I_0$ and 532-$I_2$. During daytime, Fabry-Pérot etalon filters are used (spectral widths $4 - 10$ pm). Motorized holders (FM and D3) allow for changing the day/night configuration. Motorized attenuators (A) and a motorized holder for the Iodine vapor cell (IVC) are used for calibration of the DORIS system.

DORIS system. Therefore, seeder light passes a mechanical chopper at about 2 ms after the power laser pulse and is injected into the main beam path via the back side of beam splitter BS1.

The DORIS wind system consists of the channel groups 532-$I_0$ and 532-$I_2$ and an iodine vapor cell. Beam splitter BS2 separates about 60% of the 532 nm light to be analyzed behind the $I_2$-cell. Like for the frequency stabilization of the seed laser, the iodine cell is heated and temperature stabilized to ensure that the iodine is completely in the gas phase. The cell as well

as the attenuators in front of the six DORIS channels are mounted on motorized holders for easy calibration of the DORIS system. More details about DORIS are found in Baumgarten (2010) and Hildebrand (2014).

Finally, the photons are converted to electrical signals by avalanche photo diodes (APDs) and photo multiplier tubes (PMTs). For most of the channels, APDs are used due to their much higher photon detection efficiency (e.g., ∼55% at 532 nm). Channels intended for higher altitudes are operated with electronic shutters to prevent nonlinear behavior of the detectors due to

overexposure by light at lower altitudes. This is realized by suppression of photoelectric amplification for a certain period



**Table 2.** List of channels used with the polychromatic detection system. Shown are transmitted and corresponding received wavelength ($\lambda_\text{transmitted}$, $\lambda_\text{received}$), responsible scattering process, spectral width, detector type (avalanche photo diode = APD, photomultiplier tube = PMT), electronic shutter usage and channel name. Hyphenated spectral widths indicate channels without daylight capability. Channels marked with "$I_2$ filtered" are analyzed for line of sight winds.

| $\lambda_\text{transmitted}$ | $\lambda_\text{received}$ | Scattering process | Spectral width day/night | Detector | Shutter | Channel |
|---|---|---|---|---|---|---|
| 1064 nm | 1064 nm | Rayleigh, Mie | 10 pm / 10 pm | APD | no | IH 1064 |
| | | | | APD | no | IL 1064 |
| 532 nm | 532 nm | Rayleigh, Mie | 4.6 pm / 130 pm | APD | yes | VH 532 |
| | | | | APD | yes | VM 532 |
| | | | | PMT | no | VL 532 |
| | | | | APD | yes | VDH 532 ($I_2$ filtered) |
| | | | | APD | yes | VDM 532 ($I_2$ filtered) |
| | | | | PMT | no | VDL 532 ($I_2$ filtered) |
| | 660 nm | Vib. Raman ($H_2O$) | — / 300 pm | APD | no | VVR 660 |
| | 608 nm | Vib. Raman ($N_2$) | — / 300 pm | APD | no | VVRH 608 |
| | | | | APD | no | VVRL 608 |
| | 530 nm | Rot. Raman | — / 700 pm | PMT | no | VRR 530 |
| | 529 nm | Rot. Raman | — / 700 pm | PMT | no | VRR 529 |
| 355 nm | 355 nm | Rayleigh, Mie | 4 pm / 100 pm | APD | yes | UH 355 |
| | | | | APD | yes | UM 355 |
| | | | | PMT | no | UL 355 |
| | 387 nm | Vib. Raman ($N_2$) | — / 300 pm | APD | no | UVR 387 |

after the laser pulse. The time resolved atmospheric backscatter signals are recorded by three different data acquisition (DAQ) systems, cf. Figure 1.

DAQ1 is a 250 MHz photon counting system by Licel GmbH which was installed in 2007. This rack system contains counters for 18 channels, a gating controller, high voltage supplies for PMT, and is customized for our lidar environment. The DAQ system runs on $2\times100$ Hz and can thus handle both laser/telescope systems simultaneously. It is the default data acquisition of the lidar and is usually operated at 50 m range resolution and 320 km upper altitude range. The data are stored with a time resolution of 10 s, corresponding to 1000 laser pulses per system, by hardware summation.

DAQ2 is a two channel desktop system by Licel GmbH, which is used as lidar single shot acquisition (LISA) system since 2011. It runs on $2\times33.3$ Hz and stores data of every third laser pulse with a range resolution of 25 m. The maximum altitude range of 50 km is shifted toward higher altitudes to cover the mesopause region for NLC measurements.

DAQ3 is a 500 MHz counting system realized as an embedded module featuring an FPGA and a USB3 interface controller. It operates at 200 Hz and handles 16 channels with an altitude resolution of 1.5 m to an upper altitude of 320 km. The whole





system fits on a standard eurocard sized printed circuit board. DAQ 3 stores the arrival time of photons for every laser shot with a precision of 10 ns (corresponding to 1.5 m in range). This generates a large amount of data ($\sim$ 250 GB per day in compressed
format) but has the advantage of selecting the actual resolution in time and space afterwards. DAQ 3 is our new state-of-the-art LISA system since 2017.

## 2.4 Synchronization

The trigger controller is the timing source for all synchronized operations of the lidar. It consists of an embedded controller (National Instruments sbRIO-9627) and signal conditioning electronics for digital and analog I/O, both integrated into a 19"
box similar to the laser controllers. The controller is interfaced with all essential parts of the RMR lidar, like seeder, laser transmitters, telescope control, fiber selector, polychromatic detection, and data acquisition systems. Additionally, the operation of other lidars installed in the ALOMAR observatory building can be synchronized with that of the RMR lidar. This is most significant for the tropospheric lidar because this system uses the same kind of power laser like the RMR lidar, which would cause mutual interference, especially for 532 nm channels. The laser of the tropospheric lidar runs at 33.3 Hz and an external
triggering by the RMR trigger controller. The overall timing of the three lasers is chosen in such a way that each system has finished the measurement up to its upper altitude limit before one of the lasers is transmitting the next pulse.

The RMR lidar is a distributed system containing 13 computers, including Windows and Linux desktop as well as embedded systems, which store diverse kind of data. To relate the data sets to each other, the system times need to match better than a few ms. This is realized as follows: The lidar has its own IP subnet, with a Linux computer providing the network time. Its time
source is a Meinberg GPS radio clock. All other computers synchronize their system times via the network time protocol (NTP). Additionally, timer pulses, most often the 1-second pulse by the GPS radio clock, are distributed to the embedded systems. This allows for precise data timestamps on these devices. GPS time usage for the lidar offers potential new applications, e.g. distant instruments could be synchronized with the laser pulses without direct trigger link to the lidar.

# 3 Operation

## 3.1 Security

Adherence to safety rules is especially important for systems operating high-power class-4 lasers like the RMR lidar, as direct radiation, as well as diffuse reflections, are dangerous for individuals and pose fire hazards. Other potential risky operations at the ALOMAR building are motorized tilting of heavy telescope structures and motorized opening and closing of the large aperture telescope hall. To have such risks under control, a system called ALOHA (ALOMAR Lidar Operation Health Advisory)
was developed. It consists of a number of distributed single board computers (Raspberry Pi) for monitoring and controlling infrastructure in the observatory building. Basic tasks are, e.g., door monitoring (laser and detector rooms, telescope hall, hatch access), controlling laser interlocks, hatch monitoring, and receiving data from on-site weather stations and rain radar. Follow-





ing this, the laser beams are blocked when certain doors are opened. Status displays are present on doors and other important locations.

The system was upgraded recently to allow for remote operations. For this purpose, the MQTT server plays an important role, as it collects virtually all status information of the RMR lidar and is connected to ALOHA. This allows automated rule-based decisions, like: IF (outside humidity exceeds limit) OR (rain is detected) OR (wind speed exceeds limit) OR (internet connection to remote operator is lost) THEN (put the lidar into a safe state). This standby state includes, i.e., closing the hatch of the telescope hall, closing the covers of telescope and beam guiding mirrors, stopping the data acquisition, blocking the laser

beam, and shutdown lasers by a two-step process. Additional input parameters for decisions are, i.e., fire alarm, power breaks, cloud cover, and signal quality. ALOHA supports the operator during start and stop of the lidar. Thus, this complex system can be operated by a few mouse clicks. A number of cameras are installed at the observatory building, giving visual information about the sky state in all directions, including zenith. The ALOHA system can be interfaced via web browser as well as instant messenger for mobile phones and act thus as a digital assistant for the lidar operator.

## 3.2   Laser stability

The spectral properties of seed laser as well as power laser light are monitored by several instruments, cf. Figure 2. A commercial wavelength meter (HighFiness/Ångstrom WSU-30) measures the absolute frequency of the seed laser output at 532 nm with an accuracy of 30 MHz. A custom setup consisting of an iodine vapor cell, fast photodiodes, beam splitter cubes, EOM, digital oscilloscope and software acts as a pulse spectrometer ($I_2$-Spectrometer). Part of the 532 nm outputs from the seed laser

and both power lasers is coupled into the iodine cell and is analyzed consecutively on a single-pulse basis regarding its iodine absorption. The seed laser cw light is chopped by the EOM to fit the timing given by the power lasers. This way, the frequency shifts between power lasers and seed laser are determined for every third laser pulse.

Recently, a new system was installed (E-Spectrometer). It uses an 1 GHz air spaced etalon for light analysis. Here, the interference ring patterns of light at 532 nm from the seed laser and one power laser are captured by a fast camera. The

camera images are read out by an embedded controller (National Instruments IC-3173). This compact industrial computer has similar properties to the one used for controlling the seed laser and is suitable for image processing. The seed laser cw light is synchronized by a rotating chopper, which allows a consecutive analysis of the ring patterns of seed laser and power laser with 101 Hz, i.e., every power laser pulse and one seed laser pulse per second. For each light detection, the position of the rings on the camera chip is calculated and the peak deviations of power laser pulses regarding the last measured seed laser pulse are

determined. Further details of the spectrometer and its performance will be described in a later publication.

Figure 4 contains data of 360,000 power and 3,600 seed laser pulses which were acquired during one hour. The time series in the upper panel shows a mean frequency deviation of about 15 MHz. To illustrate the necessity of a fast injection seeding control, the active tuning of the power laser oscillator length by the piezo driven mirror was stopped at about 14:10 for a duration of 1 min. (For the seeding control, the reader is referred to the last part of Section 2.1.) During this period, thermal

processes impacting the laser resonator were not compensated, which resulted in an increased frequency deviation between power and seed lasers up to 120 MHz. After restarting the control, the frequency deviations quickly returned to the former

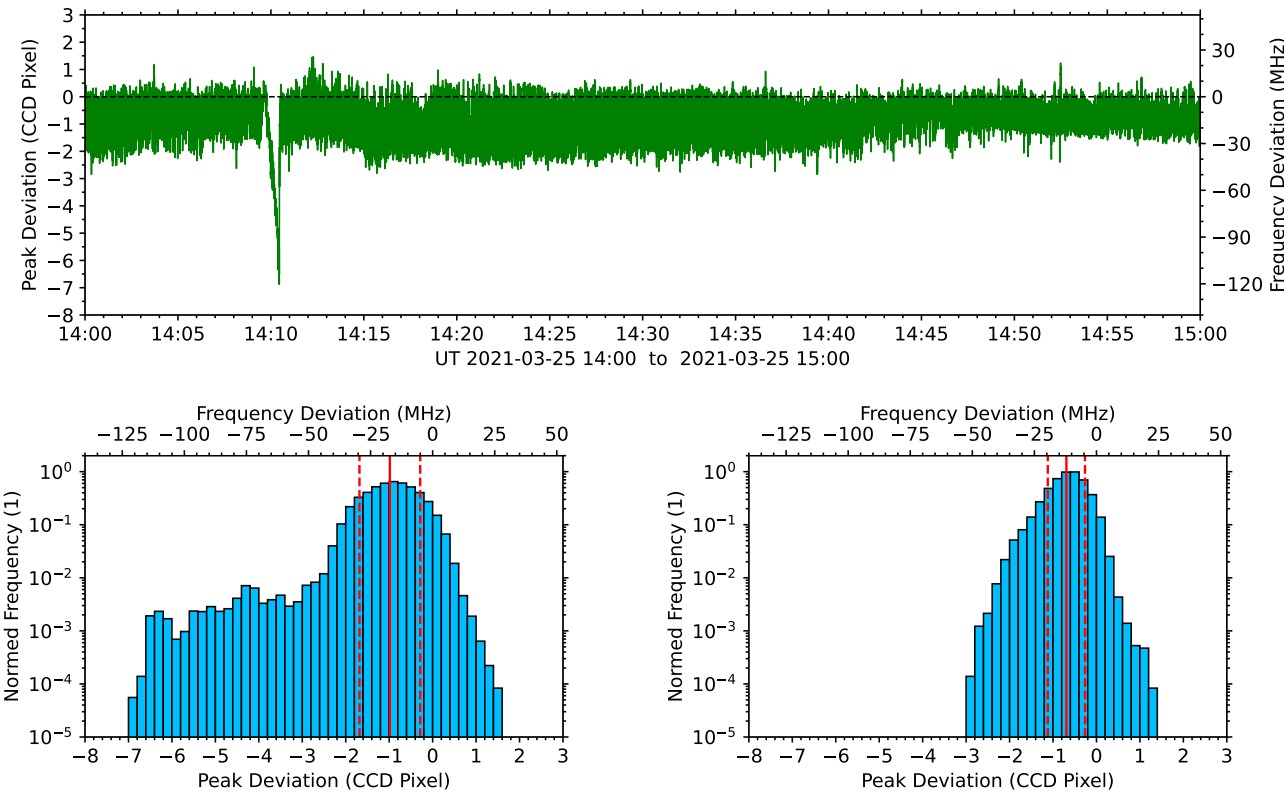

**Figure 4.** Frequency deviation between power and seed laser during 1 hour of operation. Each power laser pulse (in total 360,000 pulses) was acquired and analyzed. The active tuning of the power laser oscillator length was stopped at about 14:10 for a duration of 1 min. The upper panel shows the time series and both lower panels show histograms of frequency distributions of the deviations on a logarithmic scale. Red vertical lines indicate mean values (solid) and standard deviations (dashed). The lower left panel covers data until 14:30 and includes the period with inactive seeding control. The lower right panel covers data after 14:30, during this period the mean frequency deviation was -12±8 MHz.

level. The lower panels of Figure 4 show distributions of the frequency deviations for two periods of 30 min each, separated at 14:30. During the second period, not containing the time of inactive seeding control, a mean frequency deviation between power laser and seed laser of -12±8 MHz was obtained. The skewness of the distribution is impacted by the main thermal process (heating or cooling) of the laser resonator during the considered period.

### 3.3 Measurements

Figure 5 shows an overview of all data acquired since 1994 in terms of measurement time. For that, a minimum signal level of 0.01 counts/laserpulse/km at 50 km altitude for the high-sensitivity 532 nm channel was applied, which is a reasonable limit for calculations of geophysical parameters. Using this limit, the lidar has measured during a total of more than 20,200 hours within 30 years (lower right panel). The first year (1994) is only sparsely covered, as the lidar was installed during that summer





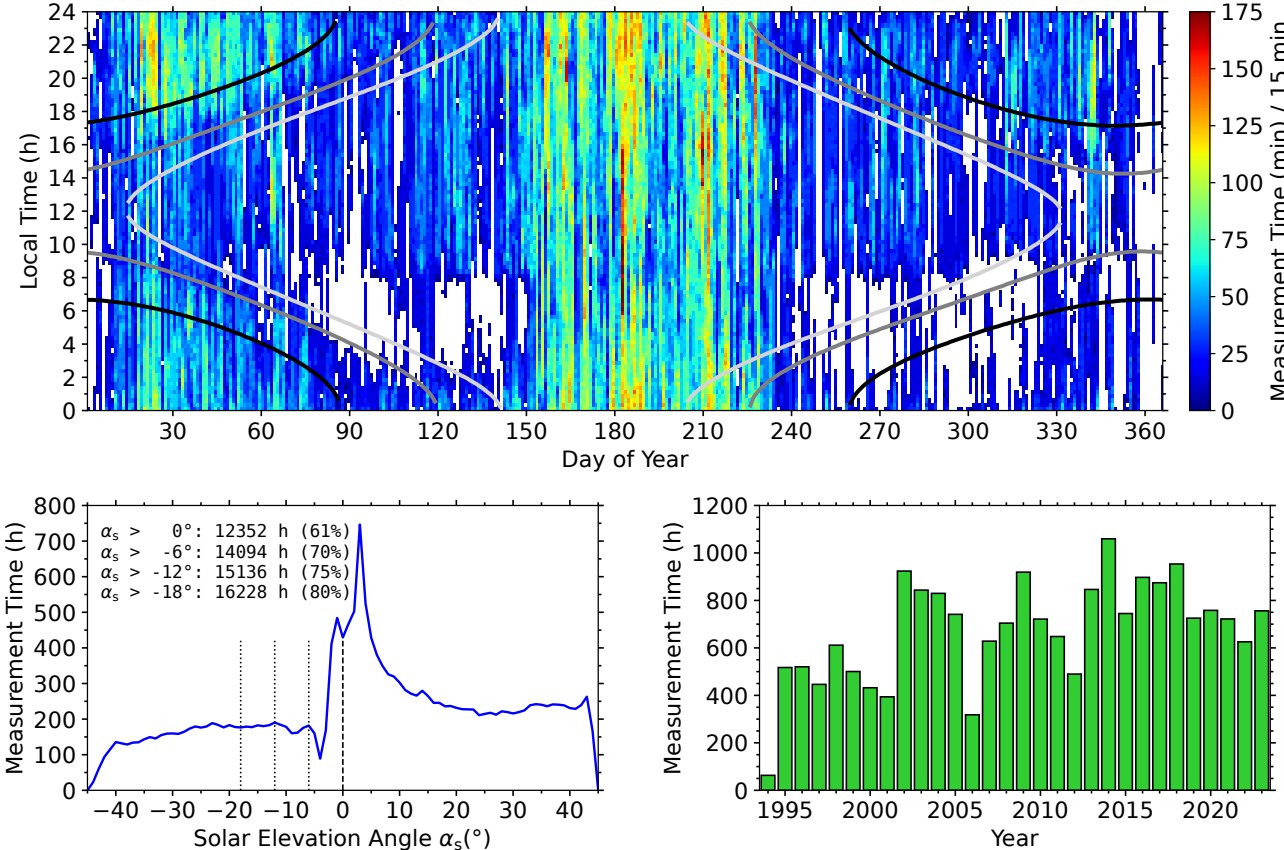

**Figure 5.** Measurement times of the ALOMAR RMR lidar from 1994 to 2023. During 30 years, more than 20,200 hours of atmospheric data were acquired. The upper panel shows the distribution of measurements over year and local time as an integrated composite. Contour lines indicate sun rise/set (white), civil twilight (gray), and astronomical twilight (black). The lower panels show the distributions of measurements over solar elevation (left) and years (right). The lower left panel additionally contains accumulated measurement times (and their percentage of total time) during certain periods of solar elevation angles, including civil ($\alpha_s = -6°$) and astronomical twilight ($\alpha_s = -18°$).

and received its first light on 19 June 1994. On average, atmospheric data were measured for ~670 hours per year. From 1995 to 2023 the actual measurement times per year vary roughly by a factor of 3 between 318 hours (2006) and 1,059 hours (2014), depending on weather conditions. These numbers correspond to an equivalent single system, most of the time the lidar was operated as a twin system, i.e., using both lasers and telescopes.

Usually, on-site Norwegian operators are in charge of the measurements during 16 hours of the working days. For weekend and nighttime operations, often students are sent to the lidar location, which is generally the case during winter and summer campaigns. However, during summer 2020 this was not possible for the first time due to travel restrictions caused by the COVID-19-pandemic. Triggered by this situation, the lidar was operated remotely from Germany during extensive times of the day. This only was made possible by finishing the ALOHA system in spring 2020. Furthermore, the travel restrictions





prevented access to the lidar for maintenance work for almost 1.5 years. Most crucial in this context would have been our previous-used power lasers due to their flashlamp exchange cycles. Fortunately, our new diode pumped power lasers came into operation in autumn 2018. Otherwise, the lidar would have stopped measurements in summer 2020, one year before the next possible visit to ALOMAR.

The upper panel of Figure 5 shows the measurement distribution over year and local time. It is an accumulated composite of 305 30 years with a time resolution of 15 minutes, resulting in a theoretical possible measurement time of 450 minutes within each 15 minute time slot. The maximum values of about 160 minutes per time slot are reached during summer and the nighttime hours in winter. Additionally, the times of several daylight conditions are shown by contour lines marking solar elevation angles of $0°$, $-6°$(civil twilight), and $-18°$(astronomical twilight). At ALOMAR the sun is continuously above the horizon for about 2 months, starting $\sim$20 May and ending ~21 July, and a maximum solar elevation angle of ~+44°is reached. During winter, 310 the sun is continuously below the horizon between ~27 November and ~14 January, with a maximum solar elevation angle of ~-3°.

The lower left panel of Figure 5 shows the distribution of measurements over solar elevation. About 60% of all measurements were performed during sunlit conditions and only 20% during complete darkness. The steep maximum around a solar elevation angle of +4°originates from the slow elevation changes during minimum conditions in summer nights.

Figure 6 shows examples of backscatter profiles during atmospheric measurements in summer as well as winter for 1 hour integration time each. The raw data were range corrected, background (solar photons and detector noise) subtracted, and filtered using running means with 500 m width. During daylight conditions, in total 11 channels, sensitive to Rayleigh and Mie scattering at the three laser wavelengths, are used (see left panel). They are equipped with etalon filters for suppression of solar photons. The summer measurement shows in the high sensitivity 532 nm channels (*VH 532*, *VDH 532*) an enhanced 320 backscatter around 83 km which is caused by mesospheric ice particles forming noctilucent clouds (NLC). Solar elevation angles were around +40°during this measurement. In darkness, the etalon filters for 532 nm and 355 nm channels are bypassed, leading to an enhanced signal level at the receivers. Additionally, channels for vibrational and rotational Raman scattering are used (see right panel), resulting in a total of 16 channels. Vibrational Raman channels are used as reference to quantify altitude ranges with aerosols, as they are insensitive to Mie scattering. This is easily visible when comparing the channels *IH 1064* 325 and *VVRH 608* on the right panel of Figure 6: the infrared channel shows an enhanced backscatter from 10 to 30 km altitude, caused by stratospheric aerosols, whereas the Raman channel does not.

## 4 Selected geophysical applications

In this section, we present examples that illustrate the performance of the RMR lidar in the investigation of basic atmospheric parameters at different altitude ranges and time scales.





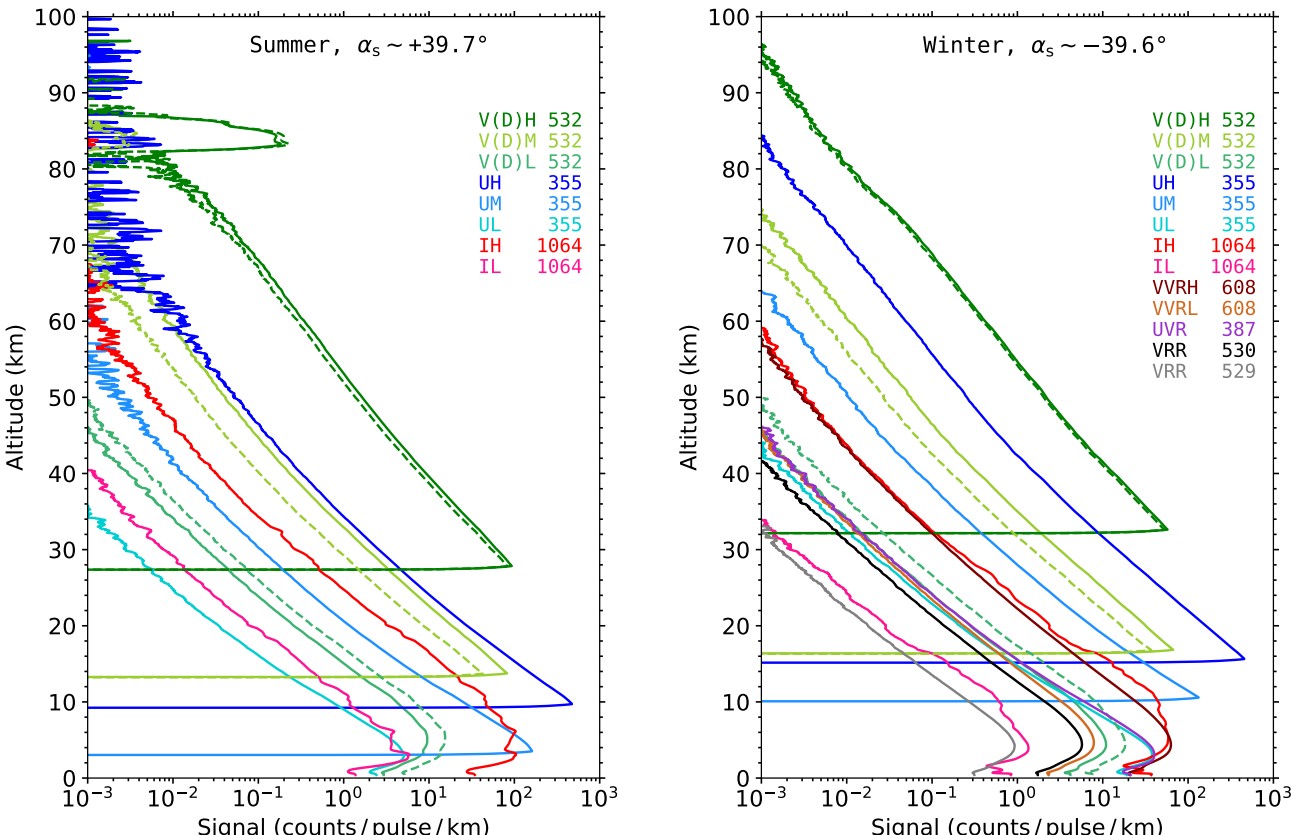

**Figure 6.** Backscatter profiles acquired at individual channels of the polychromatic detection system for summer (left panel: 2020-06-05 12:30 to 13:30 UT) and winter (right panel: 2020-01-19 00:00 to 01:00 UT). The raw signals were range corrected, background subtracted, and filtered using running means with 500 m width. Mean solar elevation angles ($\alpha_s$), channel names, and wavelengths (in nm) are indicated in the panels. During summer, channels sensitive to Rayleigh and Mie scattering are used. During winter additional channels, which are sensitive to vibrational as well as rotational Raman scattering, were used. Dotted lines show the Doppler wind channels. For details, see text.

## 4.1 Temperatures and horizontal winds

Temperature and horizontal winds are fundamental parameters for the understanding of atmospheric processes. During winter, strong westerly winds circle around cold air in the pole region at stratospheric altitudes and form the so-called stratospheric polar vortex. Due to interaction with atmospheric waves, the winds in the vortex can temporarily weaken, or even reverse in direction, which causes a rapid increase in air temperature inside the vortex. These events are named "sudden stratospheric warming" (SSW) and are important for the understanding of the wintertime stratospheric dynamics. They are marked by a reversal of the polar cap circulation, with an associated warming of several tens of degrees within a few days. During this time, the polar vortex is significantly disturbed and can be weakened, displaced, split, or even breakdown, until the winter conditions re-establish. SSW events can couple to circulation patterns in the troposphere and hence impact weather conditions.

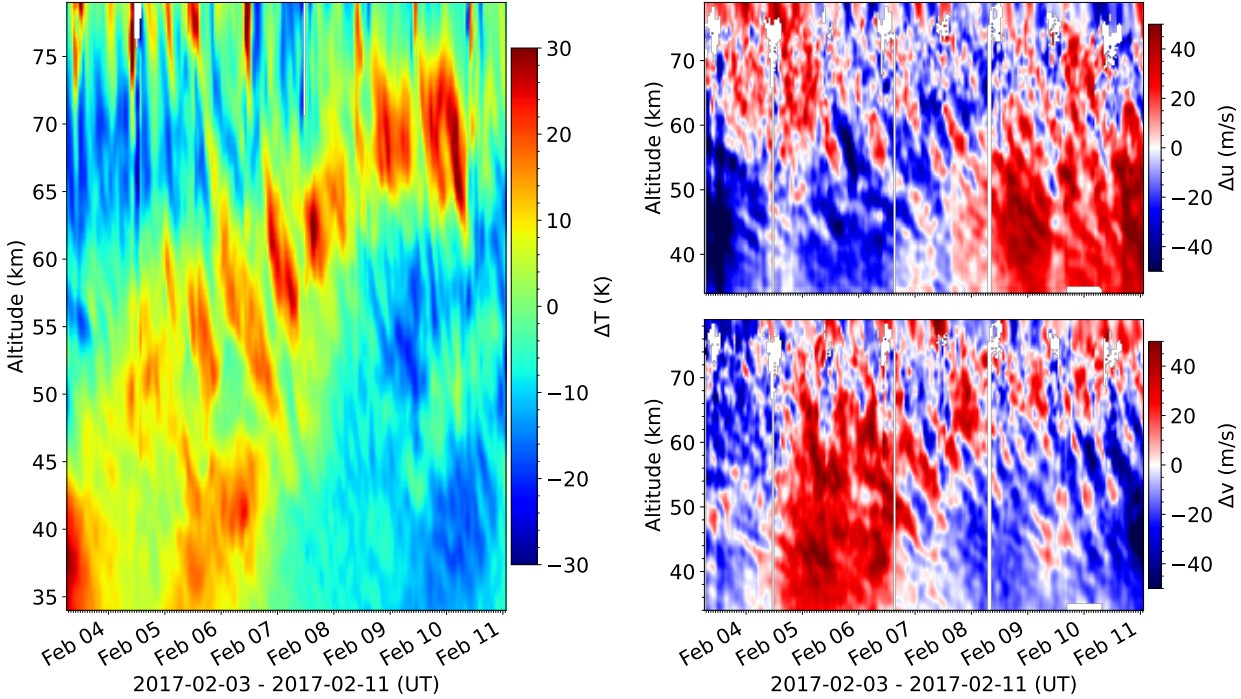

**Figure 7.** Temperature (left) and horizontal wind (right) fluctuations during a ~190 hours long measurement run in February 2017. The fluctuations are calculated by removing the average temperature and wind profiles during the measurement. Upper right panel: horizontal wind fluctuations, lower right panel: meridional wind fluctuations.

End of January 2017, a strong SSW event developed in the Northern Hemisphere, leading to a wind reversal in the strato-
sphere from westerlies to easterlies. This involved a split and shift of the polar vortex towards middle Europe, and partly Northern Germany was located inside the vortex. Starting early February, the weather conditions improved at ALOMAR and the RMR lidar documented the remaining progression of the warming event during ~190 hours of continuous and simultaneous temperature and wind measurements. Figure 7 shows temperature and horizontal wind fluctuations above ALOMAR, calcu- lated by removing the average profiles over the measurement run. The beginning of the time series shows the already descended
stratopause at an altitude of 38 km and is followed by an elevated stratopause. On top of this change of the thermal profile over several days, probably caused by planetary waves, we observe smaller scale fluctuations. The amplitudes and phases of these waves change with time and altitude, indicating a broader spectrum of gravity waves. The wave field looks rather different in the wind fluctuations, where coherent wave patterns are observed over several days with steepening wave fronts, i.e., increas- ing vertical wavelengths. Further details on the interpretation such observations is given in, e.g., Baumgarten et al. (2015) and
Strelnikova et al. (2020).





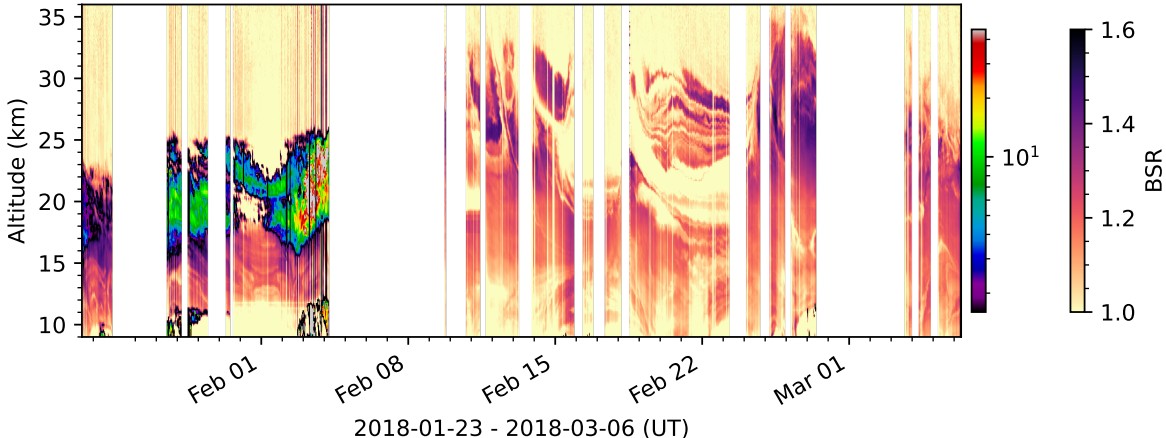

**Figure 8.** Backscatter ratio (BSR) at 1064 nm as measure for the aerosol load of the atmosphere during a period of six weeks in winter 2018. The stratospheric background aerosol is shown in sepia colors for the BSR range from 1 to 1.6. Larger BSR values are shown in rainbow colors (log-scale) and are caused by polar stratospheric clouds. Enhanced BSR up to 12 km altitude is caused by cirrus clouds.

## 4.2 Stratospheric aerosols

The lower stratosphere contains aerosols consisting of sulfuric acid / water solution droplets. These aerosols form from sulfur compounds, mainly originating from Earth's surface and volcanic eruptions. They accumulate in an altitude range from ~15 to 30 km to form a global aerosol layer around the earth. It is commonly accepted that this aerosol layer, also known as the *Junge* layer in recognition of its discoverer, has an important impact on the radiative balance of the atmosphere due to scattering of solar radiation as well as absorption of thermal radiation emitted from earth. The background aerosol load within this altitude range is modulated by large volcanic eruptions, which can increase the stratospheric sulfur burden by an order of magnitude over many months. One of these major eruptions was from Mt Pinatubo in June 1991, causing a global tropospheric temperature anomaly of -0.5° C in the following year (McCormick et al., 1995). Additionally, during winter, such aerosols serve as condensation nuclei for polar stratospheric clouds (PSC) and subsequently impact Ozone chemistry in the polar stratosphere.

Figure 8 shows the status of aerosols in the stratosphere above ALOMAR during a period of six weeks in winter 2018 as determined by scattering of the infrared wavelength of the RMR lidar at 1064 nm. The color-coded backscatter ratio (BSR) is a measure of the aerosol load, which shows distinct variations with time and altitude. Until 5 February, large BSR values between 2 and > 40 were measured, indicating the presence of PSC in the altitude range from 16 to 26 km. Enhanced backscatter up to 12 km altitude is caused by cirrus clouds at tropopause level. During the second part of the measurement period, maximum BSR values of ~1.5 were detected, which is typical for the background aerosol scattering at 1064 nm wavelength (Langenbach et al., 2019). Between 10 and 27 February, a highly dynamic state of the stratospheric aerosol layer was observed in over 300 hours of measurements. Clearly defined thin layers at roughly constant altitude lasted for several days. During the end





of February, aerosols occurred up to altitudes of 34 km. This unusual aerosol state was probably caused by a stratospheric
warming in combination with the breakdown of the polar vortex.

The backscatter ratio is usually determined as the ratio of total (aerosol and molecular) to molecular backscatter, using
elastic scattering (Rayleigh and Mie) at the laser wavelengths (1064, 532, 355 nm) and inelastic (Raman) scattering excited
by the laser wavelengths. Raman scattering is much less efficient compared to Rayleigh scattering, which results in low signal
amplitudes at stratospheric altitudes and prevents BSR calculations during daytime. To overcome this problem, Langenbach
et al. (2019) introduced a new method for BSR determinations with solely using elastic scattering and a correction function,
which was applied for the results shown in Figure 8. For a detailed view of the highly dynamic stratospheric aerosol feature
around February 22, the reader is referred to their Figure 3.

### 4.3 Mesospheric aerosols

Aerosols can also exist at exceptional high altitudes around 83 km. Doing so, they illustrate an extreme state of the Earth's
atmosphere at high geographical latitudes during summer, which is characterized by very low temperatures $< 150$ K. At these
temperatures the water vapor abundance of few ppm is sufficient to form ice particles having sizes of only few tens of nanome-
ters and number densities of about a hundred per $cm^3$ (e.g., von Cossart et al., 1999; Baumgarten et al., 2010). Such ice particles
form cloud structures showing impressive bluish-white glowing displays in the twilight sky and can be observed by the naked
eye even from mid-latitude locations. They were first documented ~140 years ago (e.g., Jesse, 1885) and given the name
noctilucent cloud (NLC). The existence of NLC particles sensitively depends on their ambient temperature and water vapor
availability and it is commonly accepted that NLC can serve as an indicator for long-term changes of the middle atmosphere
(e.g., DeLand and Thomas, 2015; Fiedler et al., 2017; Lübken et al., 2021). Additionally, NLCs show various small-scale struc-
tures in altitude and time, which are interpreted as imprints of atmospheric wave interactions (e.g., Baumgarten et al., 2009;
Baumgarten and Fritts, 2014). Such structures are found on time scales down to seconds using sophisticated lidar systems (e.g.,
Kaifler et al., 2013; Schäfer et al., 2020; Kaifler et al., 2020).

Figure 9 shows a NLC measured on July 29/30, 2022. Both telescopes were pointing vertically and their measurements have
been combined for maximum sensitivity. Data were recorded with DAQ 1 and DAQ 3 (cf. Figure 1). The example shows a
highly dynamic structure during the ∼5 hours of the NLC displayed in the upper panel of Figure 9 using a resolution of 30 s and
50 m. Altitude variations having periods of ∼45 min are clearly visible. Additionally the cloud is structured into several layers.
395  Around the end of the NLC display, the backscatter coefficient of the particles, a measure of the cloud brightness, increased
considerably. The middle panel zoomes into this area and shows a section of only 12 min with a resolution of 1 s and 30 m.
Again, the area with the largest brightness is zoomed in and shown in the lower panel, now with a resolution of 200 ms and
15 m. This panel covers a section of only 870 m and 60 s and shows cloud layering of about 100 m vertical extent. A recent
study by Schäfer et al. (2020) investigated a multi-year data set obtained by the ALOMAR RMR lidar with the result that such
400  small scale structures in NLC are not unusual.

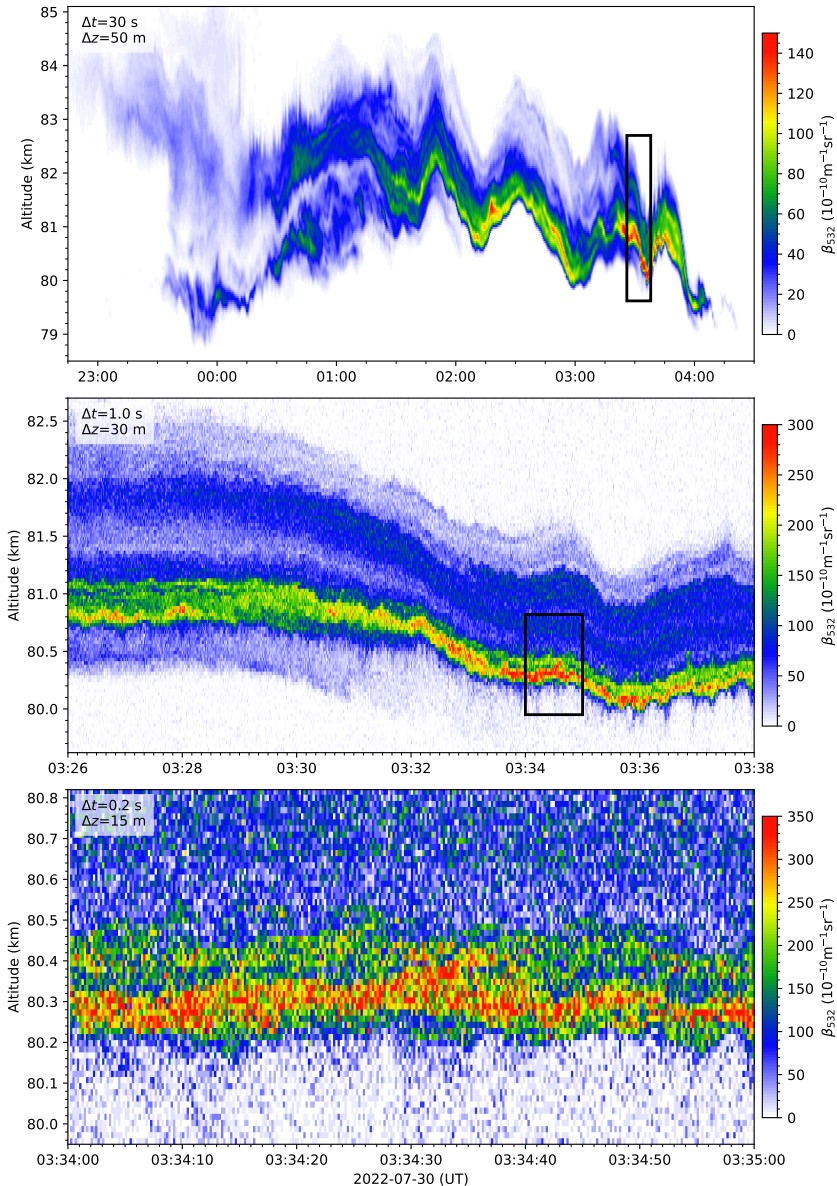

**Figure 9.** Backscatter coefficient at 532 nm wavelength as a measure of the brightness of noctilucent clouds near the edge of space. During the measurement, both telescopes were pointing vertically to maximize the signal. The upper panel shows the measurements with DAQ 1 at a resolution of 30 s and 50 m. The middle and lower panels show measurements with DAQ 3, calculated with a resolution of 1 s and 30 m (middle panel) and 200 ms and 15 m (lower panel). Please note that the color bar is changing to accommodate for the maximum brightness values observed at the different temporal and spatial scales involved. Altitude and time resolutions increase from top to bottom panels. Frames in one panel roughly indicate the altitude-time section shown in the panel below.





The time section shown in the middle panel corresponds to one data record used in our NLC statistics for investigations of long-term changes in NLC at ALOMAR (Fiedler et al., 2017). We make aware that one "statistics data point" contains a lot of fine structure.

## 5    Summary

405 The ALOMAR RMR lidar went into operation in June 1994 and acquired more than 20,200 hours of atmospheric data until the end of 2023. It was designed from the outset for multi-parameter investigations of the Arctic middle atmosphere on a climatological basis. During the past 30 years, the system was subject to constant improvements and further development with the aim of being at the cutting edge of technology. Although many components were exchanged during this process (see Table A1 for a timeline), the basic concept of this twin-lidar is unchanged: it consists of two powerful transmitters, two 410 steerable receiving telescopes, and one polychromatic detection system.

Several challenging techniques for lidars have been in routine operation for more than two decades, e.g., stable operation of high-power lasers emitting three wavelengths simultaneously, seeding of power lasers with an ultra-stable cw laser locked to a molecular absorption line, motorized guiding and direction stabilization of laser beams, motorized tilting of heavy telescope structures, usage of numerous detection channels simultaneously by spectral and intensity splitting, daylight capability at all 415 transmitted laser wavelengths, as well as computer control of all such systems.

Lately, the lidar received several major upgrades. The current power lasers (the third laser generation for the RMR lidar) allow for longer maintenance cycles due to modern diode pump technology. They have three times higher repetition rates compared to the former lasers, which demanded a new timing and trigger design for the entire lidar. The time resolution of the lidar measurements ranges from 10 s to 10 ms, depending on the used data acquisition system. Reliable Doppler wind 420 measurements are supported by spectral monitoring of each single laser pulse. Travel restrictions due to the COVID-19-pandemic prevented on-site stays for more than one year and pushed us to finish the software for remote operation of the lidar.

These days, the ALOMAR RMR lidar is just as sophisticated as it was 30 years ago, but on a much higher level. It can be operated remotely from any place in the world having an internet connection. To our best knowledge, the ALOMAR RMR 425 lidar is one of the longest operating middle atmosphere lidars and the only one measuring aerosols, temperature, and horizontal winds simultaneously and during day and night.

*Data availability.*  Lidar data in this paper is available at https://www.radar-service.eu/radar/en/dataset/hixgxNpSRPMPWPNK

## Appendix A:  ALOMAR RMR lidar timeline

The timeline of major upgrades for the ALOMAR RMR lidar is given in Table A1.



Table A1: Timeline of major upgrades for the ALOMAR RMR lidar. Text in quotation marks corresponds to notations used in Figure 1 and Figure 2.

| Year | Topic |
|------|-------|
| 1994 | Installation of lidar in the observatory building. |
| 1996 | Replacement of receiving telescope. |
|      | *new*: two 180 cm diameter mirrors (steerable); *old*: one 60 cm diameter mirror (zenith pointing) |
| 1996 | Replacement of 532 nm daylight etalon filter. |
|      | *new*: double Fabry-Pérot etalon (capacitance stabilized); *old*: single Fabry-Pérot etalon |
| 1997 | Installation of 355 nm daylight etalon filter. |
|      | *new*: double Fabry-Pérot etalon (capacitance stabilized) |
| 1998 | Installation of 1064 nm daylight etalon filter. |
|      | *new*: single Fabry-Pérot etalon (capacitance stabilized) |
| 1998 | Replacement of seed laser. |
|      | *new*: Lightwave LWE-140; *old*: Lightwave LWE-140 |
| 1999 | Installation of fiber switch at the detection system entrance. |
|      | *new*: motorized segmented mirror |
| 2002 | Replacement of seed laser. |
|      | *new*: Innolight Prometheus-30; *old*: Lightwave LWE-140 |
| 2003 | Replacement of power lasers. |
|      | *new*: Spectra-Physics PRO-290-30; *old*: Spectra-Physics GCR-6-30 |
| 2003 | Setting up of online data processing and visualization for the general public. |
|      | *active*: https://alomar.andoyaspace.no/rmrlidar/html/index.html |
| 2005 | Installation of DORIS (Doppler Rayleigh Iodine Spectrometer). |
| 2005 | Replacement of receiving telescope mirrors. |
|      | *new*: glass-ceramic body; *old*: aluminum body |
| 2005 | Replacement of seed laser. |
|      | *new*: Innolight Prometheus-50; *old*: Innolight Prometheus-30 |
| 2006 | Replacement of most photodetectors. |
|      | *new*: APD (avalanche photodiodes); *old*: PMT (photo multiplier tubes) |
| 2007 | Replacement of data acquisition system. |
|      | *new*: Licel 14 channels photon counting ["DAQ 1"]; *old*: CNRS (CAMAC) 12 channels photon counting |
| 2007 | Installation of laser pulse spectrometer. |
|      | *new*: LPS-1 |
| 2008 | Replacement of the Doppler wind system. |





| | |
|---|---|
| | *new*: DORIS (iodine gas cell); *old*: DWTS (double Fabry-Pérot etalon + ring anode imaging detector) |
| 2009 | Replacement of beam guiding mirror mounts and controllers. |
| 2010 | Replacement of 532 nm daylight etalon filter. |
| | *new*: single Fabry-Pérot etalon (pressure controlled); *old*: double Fabry-Pérot etalon (capacitance stabilized) |
| 2011 | Replacement of seed laser. |
| | *new*: Innolight Prometheus-100; *old*: Innolight Prometheus-50 |
| 2011 | Installation of data acquisition system. |
| | *new*: LISA-1 (Lidar Single Shot Acquisition) ["DAQ 2"] |
| 2013 | Replacement of laser pulse spectrometer. |
| | *new*: LPS-2 ["$I_2$-Spectrometer"] ; *old*: LPS-1 |
| 2013 | Replacement of optical fibers between telescopes and detection system. |
| | *new*: 0.8 mm diameter; *old*: 1.5 mm diameter |
| 2017 | Installation of data acquisition system. |
| | *new*: LISA-2 (Lidar Single Shot Acquisition) ["DAQ 3"] |
| 2018 | Installation of ALOHA (ALOMAR Lidar Operation and Health Administration). |
| 2018 | Replacement of power lasers. |
| | *new*: Innolas EVO IV; *old*: Spectra-Physics PRO-290-30 |
| 2018 | Replacement of lidar synchronization. |
| 2018 | Replacement of fiber switch at the detection system entrance. |
| | *new*: galvanometer scanner mirror ["RFS"]; *old*: motorized segmented mirror |
| 2019 | Replacement of telescope beam stabilization. |
| 2023 | Installation of laser pulse spectrometers. |
| | *new*: LPS-3 ["E-Spectrometer"] |

*Author contributions.* JF and GB worked on system design and construction, and participated in observations. JF prepared the manuscript with contributions from GB.

*Competing interests.* GB is member of the editorial board of Atmospheric Measurement Techniques.

*Acknowledgements.* We thank Bernd Kaifler for sharing information on the development of the power laser cavity control and the etalon housing. We thank Götz von Cossart for his excellent support in maintaining the lasers of the ALOMAR RMR lidar for many years. We thank
Torsten Köpnick and Reik Ostermann for their technical support for the lidar. We are grateful to Ulf von Zahn and Franz-Josef Lübken for





continuously supporting the ALOMAR RMR lidar during their directorship at IAP. We gratefully acknowledge the support of the ALOMAR staff in helping to accumulate the extensive data set of observations. The observations were also supported by a large number of voluntary lidar operators.



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
