# Peer review of "The ALOMAR Rayleigh/Mie/Raman lidar: status after 30 years of operation"

_Atmospheric Measurement Techniques, 2024_

## Author Comment (AC1)

We thank the reviewer for spending time and work in examining our manuscript. In the following we reply to the individual comments (in blue color).

This manuscript reports the current status of the RMR lidar at ALOMAR station. Many technical details of the subsystems (laser source, telescope and beam guiding et al.) of the lidar are introduced in this paper. This paper is well written, and will be a good reference for the experts in this field. The ability of remote operation is quite notable for the reason that to automatically operate such a lidar is quite difficult. The 30-years operation and technical upgrading makes RMR lidar a rare or even the only system in the world that can provide aerosols, temperature, and horizontal winds simultaneously day and night in the middle atmosphere. This paper is in the scope of AMT. Before final publication, some minor revision and clarification of technical details should be provided:

Line 20: typo error: "LIght"

Our intention was to indicate in capitals the origin of the acronym LIDAR. However, we have changed this as it might be confusing.

Figure 2: There is a beam monitor (BMON) module, is this for the beam direction stabilizing? If it is, the laser beam direction stabilization loop (as mentioned in line 107) should be described in detail.

Yes, this belongs to the beam direction stabilization. For the diode pumped power lasers this active stabilization is no longer necessary and the system is operated only in monitoring mode. We extended the text accordingly.

Line 165: the remaining beam fluctuations in the order of 6 to 20 urad. If the author could discuss this in more details and shown some raw camera and aligning methods (or give a reference), that would be helpful for the readers who intend to build similar systems.

We added some text and a reference which describes the initial technical realization of the stabilization system. An additional figure shows camera raw images as well as some details for finding the laser beam target position (new figure 3).

Table 1: The author only gives the beam parameters after expansion. It's better to show the parameters before expansion. The original beam parameters (divergence and pointing stability) of the Innolas Spitlight DPSS EVO-IV may be given by the factory technical report. However, did the author test them in the lidar lab? To my knowledge, the status of the laser may be changing during long term observations.

We exchanged the beam parameters with the ones before expansion (diameter measured, divergence taken from laser specs). Changing beam parameters, as experienced in the past, were mainly caused by aging of the flash lamps and are no longer a significant problem with the diode pumped lasers.

---

## Author Comment (AC2)

We would like to thank the reviewer for the very positive assessment of the ALOMAR RMR lidar and the manuscript.

This is an excellent paper. The ALOMAR lidar is one of the legends in the middle and upper atmosphere field. The authors present comprehensive descriptions of the development of the lidar system. It will be helpful to the community to understand the details of the system's hardware. The paper is well organized, and the selected measurement results are interesting and demonstrate the system's capability. Looking forward to more scientific finds come out from this system.

---

## Author Comment (AC3)

We thank Robert Sica for his very positive assessment of the ALOMAR RMR lidar and the manuscript. We appreciate his annotations in the quick report as it helped us improving the manuscript. In the following we address the comments (in blue color). Page and figure numbers refer to the original submission.

This manuscript is an excellent and thorough overview of the ALOMAR RMR developed by the authors and their team. It includes the technical details which support the numerous scientific studies cited and will be very useful to the community for understanding the details behind past and future scientific papers using measurements from the system.

Page1: We exchanged the reference for the historic cloud height measurement using a high-voltage spark with the original reference (Jones, 1949).

Page2: Station altitude is added.

Page 6: The laser cavity control by pulse build-up time reduction is referenced: Rahn, 1985.

Figure 5: The lower right panel additionally shows the number of different days with measurements per year.

Page 17: Reference to Junge, 1961

Most comments on text improvements have been taken into account, see manuscript version with changes marked.